# Design and Synthesis of Indoleamine 2,3-Dioxygenase 1 Inhibitors and Evaluation of Their Use as Anti-Tumor Agents

**DOI:** 10.3390/molecules24112124

**Published:** 2019-06-05

**Authors:** Hui Wen, Yuke Liu, Shufang Wang, Ting Wang, Gang Zhang, Xiaoguang Chen, Yan Li, Huaqing Cui, Fangfang Lai, Li Sheng

**Affiliations:** State Key Laboratory of Bioactive Substances and Function of Natural Medicine, Institute of Materia Medica, Peking Union Medical College and Chinese Academy of Medical Sciences, Beijing 100050, China; wenhui@imm.ac.cn (H.W.); liuyuke@imm.ac.cn (Y.L.); wangshufang@imm.ac.cn (S.W.); wangtingdlf@imm.ac.cn (T.W.); gzhang@imm.ac.cn (G.Z.); chxg@imm.ac.cn (X.C.); yanli@imm.ac.cn (Y.L.); hcui@imm.ac.cn (H.C.)

**Keywords:** indoleamine 2,3-dioxygenase, inhibitor, anti-tumor, immune modulation, tryptophan metabolism

## Abstract

Indoleamine 2,3-dioxygenase (IDO) 1 is the key enzyme for regulating tryptophan metabolism and is an important target for interrupting tumor immune escape. In this study, we designed four series of compounds as potential IDO1 inhibitors by attaching various fragments or ligands to indole or phenylimidazole scaffolds to improve binding to IDO1. The compounds were synthesized and their inhibitory activities against IDO1 and tryptophan 2,3-dioxygenase were evaluated. The cytotoxicities of the compounds against two tumor cell lines were also determined. Two compounds with a phenylimidazole scaffold (DX-03-12 and DX-03-13) showed potent IDO1 inhibition with IC_50_ values of 0.3–0.5 μM. These two IDO1 inhibitors showed low cell cytotoxicity, which indicated that they may exert their anti-tumor effect via immune modulation. Compound DX-03-12 was investigated further by determining the in vivo pharmacokinetic profile and anti-tumor efficacy. The pharmacokinetic study revealed that DX-03-12 had satisfactory properties in mice, with rapid absorption, moderate plasma clearance (∼36% of hepatic blood flow), acceptable half-life (∼4.6 h), and high oral bioavailability (∼96%). Daily oral administration of 60 mg/kg of compound DX-03-12 decreased tumor growth by 72.2% after 19 days in a mouse melanoma cell B16-F10 xenograft model compared with the untreated control. Moreover, there was no obvious weight loss in DX-03-12-treated mice. In conclusion, compound DX-03-12 is a potent lead compound for developing IDO1 inhibitors and anti-tumor agents.

## 1. Introduction

Tryptophan metabolism is an important pathway for cancer immunotherapy [1,2,3]. The accumulation of tryptophan can activate T cells, and a decrease in tryptophan concentration affects the regular function of T cells and interferes with the differentiation of naïve T cells. The accumulation of tryptophan metabolites, such as l-kynurenine and other downstream kynurenines (3-hydroxykynurenine and 3-hydroxyanthranilic acid), is toxic to T cells and induces T cell apoptosis. Thus, increasing tryptophan degradation inhibits the immune response [4,5,6]. 

Indoleamine 2,3-dioxygenase (IDO; E.C. 1.12.11.17) 1 is a monomeric 41 kDa enzyme that contains heme and converts l-tryptophan to *N*-formylkynurenine in non-hepatic tissues [7]. Tryptophan 2,3-dioxygenase (TDO; E.C. 1.13.11.11) also catalyzes the degradation of l-tryptophan to *N*-formylkynurenine [8]. However, these two enzymes are genetically distinct and have low sequence homology [8]. 

IDO1 is an important enzyme in tryptophan metabolism and contributes to creating peripheral immune tolerance by degrading l-tryptophan, which suppresses the function of T cells, and by producing *N*-formylkynurenine, which inhibits T cells [8]. In addition, IDO1 can also trigger other activators of anti-tumor immunity. Notably, IDO1 is overexpressed in a range of cancer tissues [4,5,6]. Thus, blocking the activity of IDO1 is a potential strategy for tumor immunotherapy.

The crystal structures of recombinant human IDO1 have been resolved [9,10], and biochemical studies have revealed that the heme iron in the active site of IDO1 binding and forms adducts with exogenous ligands including O_2_^−^, NO, CO, H_2_S, and CN^−^ [9,11,12,13,14]. IDO1 tolerates various substrates, whereas TDO has strict substrate specificity. This difference can be used to develop selective IDO1 inhibitors [8]. 

During the development of anti-tumor drugs, potent IDO1 inhibitors were discovered, and intensive efforts have been made to develop potent IDO1/TDO inhibitors with various structural scaffolds. Some typical examples of IDO1/TDO inhibitors are shown in Figure 1 [1,15,16,17,18,19,20,21]. The indole and phenylimidazole scaffolds have produced several potent IDO1 inhibitors. For example, IDO1 inhibitors indoximod (pathway IC_50_ = 450 nM) and PF-0684003 (IDO1 IC_50_ = 410 nM) are indole derivatives [13], and GDC-0919 (IDO1 IC_50_ = 13 nM) and AMCL-17g (IDO1 IC_50_ = 77 nM) are phenylimidazole derivatives [20]. 

In this study, in the early stages of searching for new IDO1 inhibitors, we used indole and phenylimidazole as the core scaffolds. Four series of compounds were designed with different strategies. The compounds were synthesized and the IDO1/TDO inhibitory activities were determined. The in vivo pharmacokinetic profile and anti-tumor efficacy of a potent IDO1 inhibitor were evaluated to explore its potential as an anti-tumor agent.

## 2. Results and Discussion

### 2.1. Design Strategy for the Four Compound Series

Four series of compounds were designed (Figure 2) to explore the structure–activity relationship (SAR). The SAR information was used for further optimization. Indole is derived from tryptophan and phenylimidazole is an IDO1 inhibitor [18]. Thus, tryptophan and phenylimidazole fit into the active site of IDO1, and they were selected as the core scaffolds for designing the inhibitors. As we mentioned in the introduction, several exogenous ligands including hydroxylamine, O_2_^−^, NO, CO, H_2_S, and CN^−^ were able to bind with the heme iron in the active site of IDO1 [9,11,12,13,14]. We attached various fragments and ligands to these scaffolds to strengthen the binding to the active site of IDO1. 

In series 1, several ligands, such as hydroxylamine and SO, that interact strongly with heme iron were linked to the double bond of the indole scaffold. We hoped to observe a synergistic effect between the ligand and the indole ring in the IDO1 binding. Series 2 was based on compound PF-0684003 (Figure 1), which is an IDO1 inhibitor developed by Pfizer. The structure of PF-0684003 co-crystalized with IDO1 revealed that there was no interaction between PF-0684003 and the heme iron, which is usually observed between other IDO1 inhibitors and IDO1 [13]. Thus, several potential heme binding ligands, such as hydroxylamine and SO, were added to the structure of PF-0684003 to enhance the inhibitory activity.

Phenylimidazole is a weak IDO1 inhibitor. Crystal structures have revealed that the imidazole ring binds with the heme iron [18,19]. Thus, in series 3 and 4, we used phenylimidazole as the scaffold, and we attached fragments to either the imidazole ring (series 3) or the phenyl ring (series 4). These fragments were intended to improve the binding in the active site of IDO1 through various interactions, such as hydrogen bonds and π–π interactions.

### 2.2. Synthesis of Selected Compounds

Although the compounds were classified into four series for examining the SAR, the structural diversity meant that they required a variety of synthetic routes. We describe representative synthetic schemes for four compounds. Syntheses for all the compounds are included in the Experimental section.

Scheme 1 shows the synthesis of DX-02-05. In the presence of a Lewis acid, ethyl 5-fluoro-1*H*-indole-2-carboxylate underwent a condensation reaction with 1*H*-pyrrole-2,5-dione to yield compound DX-02-03. Alcohol **1** was obtained by reducing DX-02-03, and then **1** was directly oxidized with MnO_2_ to produce aldehyde **2**. Aldehyde **2** was condensed with hydroxylamine hydrochloride to obtain compound DX-02-04. Finally, the double bond in DX-02-04 was reduced with NaBH_3_CN to obtain compound DX-02-05.

Scheme 2 shows the synthesis of DX-02-07. Sulfur substitution of 5-fluoroindolin-2-one was performed with P_2_S_5_ to produce compound **3**. Under basic conditions, compound **4** was obtained through an SN_2_ substitution reaction with CH_3_I. A condensation reaction between **4** and 1*H*-pyrrole-2,5-dione was performed at 120 °C to obtain sulfide DX-02-06, which was oxidized to racemic sulfoxide DX-02-07.

Scheme 3 shows the synthesis of DX-03-12. First, the imidazole starting material was protected with trityl chloride to avoid various side reactions. Compound **6** was obtained by the Suzuki–Miyaura reaction with a boronic acid [22]. The methyl group was removed by BBr_3_ to expose the hydroxyl group in compound **7**. Compound **8** was prepared in parallel by melting 3-chloropropylamine and isobenzofuran-1,3-dione together. Compounds **7** and **8** were subjected to an SN_2_ nucleophilic reaction to obtain compound **9** under strong basic conditions. This step had a low yield, which was explained by the high steric hindrance of the large molecules and the weak nucleophilicity of compound **7**. The amine protecting group was removed with hydrazine hydrate to obtain compound **10**. Compound **10** was reacted with trifluoromethanesulfonylchloride under basic conditions to give compound **11**. Finally, DX-03-12 was obtained by removing the protecting group on the imidazole with acetic acid.

Scheme 4 shows the synthesis of DX-04-02. 2-Iodoimidazole was substituted with trifluoromethylbenzyl bromide to give compound **12**. Compound **12** underwent a Suzuki–Miyaura coupling reaction with (5-chloro-2-methoxyphenyl)boronic acid to obtain DX-04-01. Finally, the methyl group was removed with BBr_3_ to obtain DX-04-02.

### 2.3. In Vitro Biological Evaluation

We prepared the four series of compounds using the synthetic schemes in Section 2.2. Twelve commercially available compounds were purchased (see Section 3.1). In total, we screened 50 compounds in vitro for their IDO1 and TDO inhibitory activities and cell cytotoxicities against tumor cell lines H460 and MCF7 (Table 1). NLG919 was the reported potent IDO1/TDO inhibitor, and Taxol are known commercial anti-cancer drug. They were used as the reference compounds in this study.

The IDO1 and TDO inhibitory activities of the 22 compounds in series 1 were measured (Table 1). The series 1 compounds contained heme binding fragments including CO, NO, CN, H_2_S, imidazole, and hydroxylamine; however, the inhibition data showed that most compounds in series 1 exhibited marginal IDO1/TDO inhibition. However, four compounds (DX-01-14, DX-01-15, DX-01-16, and DX-01-17) containing hydroxylamine had moderate IDO1 inhibitory activities. The most active IDO1 inhibitor in series 1 was compound DX-01-14 with an IC_50_ of 11.1 μM. In addition, cell cytotoxicity screening showed that most of the compounds were not toxic to mammalian cancer cell lines. 

Generally speaking, these molecules have a rather low molecular weight, and the rather small moiety causes them hard to bind tightly inside the active center of IDO1. Thus, a poor SAR was observed for this series of compounds. Interestingly, for these four active compounds, they all contain the fragment of hydroxylamine. This may suggest that hydroxylamine is the ideal ligand to attach to the double bond of the indole scaffold, while still maintaining the binding with the heme iron. 

Series 2 included seven compounds, which were all PF-0684003 derivatives (Table 2). However, the in vitro IDO1/TDO inhibitory activities of all these PF-0684003 derivatives were lower than that of the parent compound, and none were IDO1/TDO inhibitors. The cytotoxicity study demonstrated that these PF-0684003 derivatives were not toxic to cancer cell lines.

The structure of PF-0684003 co-crystallized with IDO1 shows that PF-0684003 binds tightly in the active site of IDO1 but does not interact with the heme iron [18]. In the present study, the addition of the heme binding ligands [14] (SO and hydroxylamine in DX-02-07 and DX-02-05, respectively) to the double bond of the indole ring abrogated the IDO1 inhibition, similar to the optimization reported in a previous study.

Series 3 contains 13 phenylimidazole derivatives (Table 3). Compounds DX-03-01 to DX-03-10 showed marginal IDO1/TDO inhibition and weak cell cytotoxicities. However, compounds DX-03-11, DX-03-12, and DX-03-13 are potent IDO1 inhibitors with IC_50_ values of 0.3–2.4 μM. Interestingly, these three compounds show marginal TDO inhibition activity, and thus are IDO1/TDO selective inhibitors. In addition, these compounds are not toxic to mammalian cancer cell lines.

Compound DX-03-01 is a known compound [18], which binds to the active site in IDO1. The imidazole ring interacts with the heme iron, and the hydroxyl group binds to the nearby Ser167 in the active site [7]. However, the substitution of the hydroxyl group with hydroxylamine (DX-03-01 to DX-03-10) did not produce IDO1 inhibitors with higher activity. In DX-03-10 to DX-03-13, the hydroxyl group was used to attach an alkyl chain bearing terminal groups used in other reported IDO1 inhibitors [17,19]. The terminal fragments, such as sulfamide and urea, are important to maintain the binding [17,19]. 

Series 4 contains eight phenylimidazole derivatives (Table 4). Most compounds in this series showed marginal IDO1 inhibition, but compound DX-04-05 showed moderate IDO1 inhibition. However, compound DX-04-06, in which the hydroxyl group was methylated, showed decreased IDO1 inhibition. Most compounds in this series were toxic to the cancer cell lines H460 and MCF7 with an IC_50_ of around 10^−5^ M, which indicates the compounds also interrupt the function of other targets in the cells. 

In series 4, the design strategy was to attach fragments to the imidazole ring instead of the phenyl ring in series 3. However, this strategy does not maintain the IDO1 inhibitory activity. In addition, the free hydroxyl group in series 4 appears to be important for forming a hydrogen bond with an amino acid residue, such as Ser167 [7], in the active site; methylating this hydroxyl group decreased IDO1 inhibition (DX-04-05 versus DX-04-06).

In summary, based on the SAR study, compounds DX-03-12 and DX-03-13 showed potent IDO1 inhibition activity, and compound DX-03-13 has some structure similarity as previous published patent (WO2011/056652Al) [23]. Thus, the most potent IDO1 inhibitor DX-03-12 was used as the lead compound for further study.

### 2.4. Pharmacokinetic Study of DX-03-12

The in vivo pharmacokinetic properties of compound DX-03-12 were evaluated in male ICR mice. DX-03-12 was administered orally either as a bolus at a dose of 30 mg/kg or intravenously at a dose of 3 mg/kg. The plasma concentration-time profiles are shown in Figure 3. The pharmacokinetic parameters were determined by non-compartmental analysis (Table 5). After intravenous administration (3 mg/kg), compound DX-03-12 showed a moderate clearance of 33.3 mL/min/kg, which was approximately 36% of mouse hepatic blood flow (90 mL/min/kg). The volume of distribution in the steady state was 10 L/kg, suggesting that there was an extensive distribution of compound DX-03-12 in tissues. Compound DX-03-12 was rapidly absorbed after oral administration of 30 mg/kg. The plasma concentration was quantifiable at the first sampling time point (5 min) and remained detectable at 24 h. The maximum plasma concentration of compound DX-03-12 of 2963 ng/mL was observed 1 h post-dose. Compound DX-03-12 was eliminated relatively slowly with a half-life of 4.6 h. The oral bioavailability of compound DX-03-12 was calculated to be 96%.

The pharmacokinetic study revealed that compound DX-03-12 showed rapid absorption and almost complete oral bioavailability in mice. After oral administration of 30 mg/kg in mice, it exhibited a long half-life (4.6 h) due to moderate plasma clearance. The pharmacokinetic properties of compound DX-03-12 in mice indicate that it has the potential to be a once-a-day drug.

### 2.5. In Vivo Anti-Tumor Efficacy Study of Compound DX-03-12

To determine whether the compound was effective against tumors in vivo, the anti-tumor effect of DX-03-12 was evaluated in a B16F10 subcutaneous xenograft model in syngeneic mice (Figure 4). Oral and intraperitoneal (i.p.) administration (30 or 60 mg/kg/day) of DX-03-12 significantly decreased the growth of melanoma B16F10 xenograft tumors in a dose-dependent manner. DX-03-12 oral treatment resulted in a 72.2% decrease in tumor weight and i.p. treatment resulted in a 72.3% decrease at a dose of 60 mg/kg compared with the control tumor after 19 days treatment. DX-03-12 had no obvious effect on the body weight of the mice and the peripheral white blood cells, which indicated that DX-03-12 had low toxicity in mice. In this study, the classic first line anti-cancer drug cyclophosphamide (CTX) was used as the reference drug.

## 3. Experiment Section

### 3.1. Chemistry

Reagents and solvents were obtained from commercial suppliers and used as received. ^1^H-NMR spectra were obtained on an NMR spectrometer (Mercury, Varian, San Diego, CA, USA; 400 MHz). Electrospray ionization (ESI) mass spectra and high-resolution mass spectroscopy (HRMS) were performed with a liquid chromatograph/mass selective detector time-of-flight mass spectrometer (LC/MSD TOF, Agilent Technologies, Santa Clara, CA, USA). silica gel column chromatography was performed with silica gel 60G (Qingdao Haiyang Chemical, Qingdao, China). Purity was determined using HPLC, LC/MS and NMR spectroscopy. All of the synthesized compounds have the purity over than 95%.

Several commercial available compounds were purchased from Beijing innochem Co. Ltd. (Beijing, China). They are DX-01-01, DX-01-08, DX-01-09, DX-01-10, DX-01-11, DX-01-12, DX-01-18, DX-01-19, DX-01-20, DX-01-21, DX-01-22, DX-03-01.

#### 3.1.1. Preparation of (*R/S*) 3-(5-fluoro-2-((hydroxyamino)methyl)-1*H*-indol-3-yl)pyrrolidine-2,5-dione (DX-02-05)

##### Preparation of (*R/S*) Ethyl 3-(2,5-dioxopyrrolidin-3-yl)-5-fluoro-1*H*-indole-2-carboxylate (DX-02-03)

Ethyl 5-fluoro-1*H*-indole-2-carboxylate (2 g, 9.7 mmol) and 1*H*-pyrrole-2,5-dione (1.40 g, 14.4 mmol) were added to dry 1,2-dichloroethane (50 mL), and 46.5% BF_3_-Et_2_O (3.54 g, 11.6 mmol) was added to reaction mixture at 20 °C. After the reaction mixture was stirred at 90 °C, until the starting material disappeared in thin layer chromatography (TLC), the reaction mixture was distilled in vacuo, and dichloromethane (50 mL) was added to the mixture. The organic extract was washed twice with saturated aqueous NaHCO_3_ and NaCl (20 mL) respectively, and the organic extract was dried over Na_2_SO_4_. After solvent was removed under vacuum, the product was purified as a white solid by silica gel column chromatography, 1.62 g, yield 55%. ^1^H-NMR (400 MHz, DMSO-*d*_6_): δ = 12.00 (s, 1H, NH-indolyl), 11.31 (s, 1H, NH), 7.49–7.45 (m, 2H, H-indolyl), 7.18 (td, *J* = 9.2, 2.6 Hz, 1H, H-indolyl), 4.82 (dd, *J* = 9.6, 6.6 Hz, 1H, CH), 4.33–4.27 (m, 2H, CH_2_), 3.08 (dd, *J* = 17.7, 9.6 Hz, 1H, CH’H’’), 2.70 (dd, *J* = 17.7, 6.6 Hz, 1H, CH’H’’), 1.30 (t, *J* = 7.1 Hz, 3H, CH_3_). ^13^C-NMR (101 MHz, CD_3_OD): δ =182.17, 180.15, 162.66, 160.66, 134.24, 129.01, 126.54, 119.11, 115.66, 114.82, 104.78, 62.19, 39.68, 39.37, 14.63. HRMS (ESI): *m/z* [M + H]^+^ calculated for C_15_H_14_O_4_N_2_F: 305.09321; found: 305.09290.

##### Preparation of (*R/S*) 3-(5-fluoro-2-(hydroxymethyl)-1*H*-indol-3-yl) pyrrolidine-2,5-dione (**1**)

Ethyl 3-(2,5-dioxopyrrolidin-3-yl)-5-fluoro-1*H*-indole-2-carboxylate (DX-02-03, 1.3 g, 4.3 mmol) were added to dry CH_3_OH (20 mL), LiAlH_4_ (0.13 g, 4.3 mmol) was added to reaction mixture gradually at 20 °C. The reaction mixture was stirred at 20 °C until the starting material disappeared in TLC. The reaction mixture was distilled in vacuo, and ethyl acetate (50 mL) was added to the mixture. The organic extract was washed twice with saturated aqueous NaCl (20 mL), and the organic extract was dried over Na_2_SO_4_. After solvent was removed under vacuum, the product was purified as a white solid by silica gel column chromatography, 0.97 g, yield 86%. ^1^H-NMR (400 MHz, CD_3_OD): δ = 7.30 (dd, *J* = 8.8, 4.4 Hz, 1H, H-phenyl), 6.93 (dd, *J* = 9.9, 2.4 Hz, 1H, H-phenyl), 6.89–6.83 (m, 1H, H-phenyl), 4.72 (d, *J* = 1.3 Hz, 2H, CH_2_), 4.46 (dd, *J* = 9.7, 5.6 Hz, 1H, CH), 3.20 (dd, *J* = 18.4, 9.7 Hz, 1H, CH’H’’), 2.81 (dd, *J* = 18.4, 5.6 Hz, 1H, CH’H’’). ^13^C-NMR (101 MHz, CD_3_OD): δ = 182.67, 180.16, 160.22, 157.85, 139.57, 133.85, 127.82, 113.46, 110.70, 103.45, 56.67, 39.93, 39.00. HRMS (ESI): *m/z* [M + H]^+^ calculated for C_13_H_12_N_2_O_3_F: 263.08265; found: 263.08160.

##### Preparation of (*R/S*) 3-(2,5-dioxopyrrolidin-3-yl)-5-fluoro-1*H*-indole-2-carbaldehyde (**2**)

3-(5-Fluoro-2-(hydroxymethyl)-1*H*-indol-3-yl)pyrrolidine-2,5-dione (**1**, 0.8 g, 3.1 mmol) was added to dry dichloridemethane (20 mL), MnO_2_ (0.36 g, 4.1 mmol) was added to reaction mixture gradually at 20 °C. The reaction mixture was stirred at 20 °C until the starting material disappeared in TLC. The reaction mixture was filtered, the solvent was removed under vacuum, and the product was purified as a white solid by silica gel column chromatography, 0.73g, yield 91%. ^1^H-NMR (400 MHz, CD_3_OD): δ = 10.00 (s, 1H, CHO), 7.50 (dd, *J* = 9.1, 4.3 Hz, 1H, H-phenyl), 7.27 (dd, *J* = 9.5, 2.3 Hz, 1H, H-phenyl), 7.19 (td, *J* = 9.1, 2.4 Hz, 1H, H-phenyl), 4.91 (dd, *J* = 9.7, 6.0 Hz, 1H, CH), 3.29–3.21 (m, 1H, CH’H’’), 2.84 (dd, *J* = 18.1, 6.0 Hz, 1H, CH’H’’). ^13^C-NMR (101 MHz, CD_3_OD): δ = 183.66, 181.23, 179.66, 160.59, 158.30, 135.75, 135.36, 127.61, 117.26, 115.66, 105.59, 39.77, 39.35. HRMS (ESI): *m/z* [M + H]^+^ calculated for C_13_H_10_N_2_O_3_F: 261.06700; found: 261.06638.

##### Preparation of (*R/S*) (*E*)-3-(2,5-dioxopyrrolidin-3-yl)-5-fluoro-1*H*-indole-2-carbaldehyde oxime (DX-02-04)

3-(2,5-Dioxopyrrolidin-3-yl)-5-fluoro-1*H*-indole-2-carbaldehyde (0.5 g, 1.92 mmol) was added to dry CH_3_OH (30 mL), then *N,N*-Diisopropylethylamine (0.5 g, 3.84 mmol) and hydroxylamine hydro -chloride (0.13 g, 1.92 mmol) were added respectively at 20 °C. The reaction mixture was stirred at 60 °C until the starting material disappeared in TLC. After solvent was removed under vacuum, the product was purified as a white solid by silica gel column chromatography, 0.35 g, yield 66%. ^1^H-NMR (400 MHz, CD_3_OD): δ = 8.22 (s, 1H, =CH), 7.32 (dd, *J* = 8.8, 4.4 Hz, 1H, H-indolyl), 7.06–6.86 (m, 2H, H-indolyl), 4.63 (dd, *J* = 9.7, 5.9 Hz, 1H, CH), 3.19 (dd, *J* = 18.3, 9.7 Hz, 1H, CH’H’’), 2.81 (dd, *J* = 18.7, 6.2 Hz, 1H, CH’H’’). ^13^C-NMR (101 MHz, CD_3_OD): δ =181.92, 179.89, 160.31, 157.98, 141.03, 136.51, 134.81, 132.30, 113.78, 112.58, 103.95, 39.83, 38.66. HRMS (ESI): *m/z* [M + H]^+^ calculated for C_13_H_11_N_3_O_3_F: 276.07790; found: 276.07718.

##### Preparation of (*R/S*) 3-(5-fluoro-2-((hydroxyamino)methyl)-1*H*-indol-3-yl) pyrrolidine-2,5-dione (DX-02-05)

3-(2,5-Dioxopyrrolidin-3-yl)-5-fluoro-1*H*-indole-2-carbaldehyde oxime (DX-02-04, 0.3 g, 1.1 mmol) was added to CH_3_OH (20 mL), then NaBH_3_CN (0.07 g, 1.1 mmol) was added at 20 °C. 12N HCl was added to reaction mixture continuously to keep acidic. The reaction mixture was stirred at 20 °C until the starting material disappeared in TLC. After solvent was removed under vacuum, the product was purified as a white solid by silica gel column chromatography, 0.13 g, yield 43%. ^1^H-NMR (400 MHz, CD_3_OD): δ = 7.30 (dd, *J* = 8.8, 4.5 Hz, 1H, H-indolyl), 6.93–6.88 (m, 1H, H-indolyl), 6.88–6.82 (m, 1H, H-indolyl), 4.50 (dd, *J* = 9.7, 5.5 Hz, 1H, CH), 4.12 (d, *J* = 3.6 Hz, 2H, CH_2_NH), 3.21 (dd, *J* = 18.5, 9.7 Hz, 1H, CH’H’’), 2.85 (dd, *J* = 18.5, 5.5 Hz, 1H, CH’H’’). ^13^C-NMR (101 MHz, CD_3_OD): δ = 160.27, 157.92, 135.20, 130.85, 130.02, 113.00, 110.85, 110.54, 104.55, 104.31, 50.48, 38.34, 30.37. HRMS (ESI): *m/z* [M + H]^+^ calculated for C_13_H_13_N_3_O_3_F: 278.09355; found: 278.09291.

#### 3.1.2. Preparation of (*R/S*) 3-(5-fluoro-2-(methylsulfinyl)-1*H*-indol-3-yl)pyrrolidine-2,5-dione (DX-02-07)

##### Preparation of 5-fluoroindoline-2-thione (**3**)

5-fluoroindolin-2-one (2 g, 13 mmol) and P_2_S_5_ (2.92 g, 13 mmol) were added to dry tetrahydrofuran (30 mL). After the reaction mixture was stirred at room temperature for 1 h. Until the starting material disappeared in thin layer chromatography (TLC), the NaHCO_3_ (3.43 g, 39 mmol) was added gradually. After stirring at room temperature for 1 h, the reaction mixture was distilled in vacuo, dichloromethane (50 mL) was added to the mixture. The organic extract was washed twice with saturated aqueous NaCl (20 mL), and the organic extract was dried over Na_2_SO_4_. After solvent was removed under vacuum, the product was obtained as yellow crystals by re-crystallization using ethanol, 1.62 g, yield 75%. ^1^H-NMR (400 MHz, CDCl_3_): δ = 10.14 (s, 1H, NH), 7.05–6.95 (m, 1H, H-phenyl), 6.94–6.89 (m, 1H, H-phenyl), 4.08 (s, 1H, CH_2_). ^13^C-NMR (101 MHz, CDCl_3_)): δ = 203.39, 159.10, 140.24, 132.15, 114.87, 112.47, 110.47, 49.15. HRMS (ESI): *m/z* [M + H]^+^ calculated for C_8_H_7_NFS: 168.02777; found: 168.02773.

##### Preparation of 5-fluoro-2-(methylthio)-1*H*-indole (**4**)

5-fluoroindoline-2-thione (**3**, 1 g, 6 mmol), CH_3_I (0.99 g, 7 mmol) and K_2_CO_3_ (0.97 g, 7 mmol) were added to acetone (30 mL), and the reaction mixture was stirred at room temperature until the starting material disappeared in TLC. The reaction mixture was distilled in vacuo, ethyl acetate (50 mL) was added to the mixture. The organic extract was washed twice with saturated aqueous NaCl (20 mL), and the organic extract was dried over Na_2_SO_4_. After solvent was removed under vacuum, the product was purified as a white solid by silica gel column chromatography, 0.90 g, yield 83%. ^1^H-NMR (400 MHz, CDCl_3_): δ = 8.04 (s, 1H, NH), 7.23–7.13 (m, 2H, H-indolyl), 6.91 (s, 1H, H-indolyl), 6.49 (s, 1H, H-indolyl), 2.52 (s, 3H, CH_3_). ^13^C-NMR (101 MHz, CDCl_3_): δ = 159.18, 156.85, 133.57, 129.02, 110.94, 110.47, 105.31, 104.73, 18.86. HRMS (ESI): *m/z* [M + H]^+^ calculated for C_9_H_9_NFS: 182.04342; found: 182.04425.

##### Preparation of (*R/S*) 3-(5-fluoro-2-(methylthio)-1*H*-indol-3-yl)pyrrolidine-2,5-dione (DX-02-06)

5-Fluoro-2-(methylthio)-1*H*-indole (**4**, 0.5 g, 2.7 mmol) and 1*H*-pyrrole-2,5-dione (0.54 g, 5.4 mmol) were added to dry CH_3_CO_2_H (30 mL), and the reaction mixture was stirred at 120 °C until the starting material disappeared in TLC. The reaction mixture was distilled in vacuo, ethyl acetate (100 mL) was added to the mixture. The organic extract was washed twice with saturated aqueous NaCl (30 mL), and the organic extract was dried over Na_2_SO_4_. After solvent was removed under vacuum, the product was purified as a white solid by silica gel column chromatography, 0.47 g, yield 61%. ^1^H-NMR (400MHz, CD_3_OD): δ = 7.28 (dd, *J* = 8.9, 4.5 Hz, 1H, H-phenyl), 7.00 (dd, *J* = 9.7, 2.4 Hz, 1H, H-phenyl), 6.91 (td, *J* = 9.2, 2.5 Hz, 1H, H-phenyl), 4.56 (dd, *J* = 9.8, 5.6 Hz, 1H, CH), 3.22 (dd, *J* = 18.3, 9.8 Hz, 1H, CH’H’’), 2.77 (dd, *J* = 18.3, 5.6 Hz, 1H, CH’H’’), 2.41 (s, 3H, CH_3_). ^13^C-NMR (101 MHz, CD_3_OD): δ = 182.38, 180.11, 160.26, 135.05, 132.13, 127.67, 116.34, 113.18, 111.70, 103.25, 40.35, 39.10, 19.59. HRMS (ESI): *m/z* [M + H]^+^ calculated for C_13_H_12_O_2_N_2_FS: 279.05980; found: 279.05930.

##### Preparation of (*R/S*) 3-(5-fluoro-2-(methylsulfinyl)-1*H*-indol-3-yl)pyrrolidine-2,5-dione (DX-02-07)

3-(5-Fluoro-2-(methylthio)-1*H*-indol-3-yl)pyrrolidine-2,5-dione (DX-02-06, 0.2 g, 0.7 mmol) was added to dry dichloridemethane (20 mL), and 3-Chloroperoxybenzoic acid (0.15 g, 0.9 mmol) was added gradually at 0 °C. The reaction mixture was stirred at 0 °C until the starting material disappeared in TLC, and Na_2_S_2_O_3_ (0.11 g, 0.7 mmol) was added to the mixture. After reaction mixture was stirred for 20 min, the reaction mixture was washed twice with saturated aqueous NaCl (20 mL), and the organic extract was dried over Na_2_SO_4_. After solvent was removed under vacuum, the product was purified as a white solid by silica gel column chromatography, 0.074 g, yield 35%. ^1^H-NMR (400MHz, DMSO-*d*_6_): δ = 12.28 (s, 1H, NH), 11.51 (s, 1H, NH), 7.51–7.48 (m, 1H, H-phenyl), 7.19–7.13 (m, 2H, H-phenyl), 4.61 (ddd, *J* = 26.8, 9.7, 5.8 Hz, 1H, CH), 3.18 (ddd, *J* = 18.0, 9.7, 2.3 Hz, 1H, CH’H’’), 2.98 (d, *J* = 16.7 Hz, 3H, CH_3_), 2.76 (ddd, *J* = 41.8, 18.1, 5.8 Hz, 1H, CH’H’’). ^13^C-NMR (101 MHz, DMSO-d_6_): δ = 179.74, 177.94, 158.72, 138.29, 134.14, 125.83, 114.50, 113.93, 113.53, 104.46, 41.67, 38.62, 38.11. HRMS (ESI): *m/z* [M + H]^+^ calculated for C_13_H_12_O_3_N_2_FS: 295.05472; found: 295.05508.

#### 3.1.3. Preparation of *N*-(3-(2-(1*H*-imidazol-5-yl)phenoxy)propyl)-1,1,1-trifluoromethane sulfonamide (DX-03-12)

##### Preparation of 4-iodo-1-trityl-1*H*-imidazole (**5**)

4-Iodo-1*H*-imidazole (5 g, 25.8 mmol) and *N,N*-diisopropylethylamine (6.65 g, 51.5 mmol) were added to dry dimethylformamide (30 mL), and (chloromethanetriyl)tribenzene (7.89 g, 28.4 mmol) was added to reaction mixture gradually at 20 °C. After the reaction mixture was stirred at 20 °C until the starting material disappeared in thin layer chromatography (TLC), the reaction mixture was distilled in vacuo, ethyl acetate (200 mL) was added to the mixture. The organic extract was washed twice with saturated aqueous NaCl (40 mL), and the organic extract was dried over Na_2_SO_4_. After solvent was removed under vacuum, the product was purified as a white solid by silica gel column chromatography, 8.77 g, yield 78%. ^1^H-NMR (400 MHz, CDCl_3_): δ = 7.38–7.33 (m, 10H, H-phenyl, H-imidazolyl), 7.13–7.09 (m, 6H, H-phenyl), 6.92 (d, *J* = 1.4 Hz, 1H, H-imidazolyl). ^13^C-NMR (101 MHz, CDCl_3_): δ = 141.83, 140.56, 129.72 (6C), 128.29 (3C), 128.18 (6C), 127.91, 126.89 (3C), 75.84. HRMS (ESI): *m/z* [M + H]^+^ calculated for C_22_H_18_N_2_I: 437.05092; found: 437.04934.

##### Preparation of 4-(2-methoxyphenyl)-1-trityl-1*H*-imidazole (**6**)

4-Iodo-1-trityl-1*H*-imidazole (**5**, 2 g, 4.58 mmol), (5-chloro-2-methoxyphenyl)boronic acid (0.852 g, 4.58 mmol), K_3_PO_4_ (2.9 g, 13.7 mmol), Pd(PPh_3_)_4_ (0.74 g, 0.64 mmol) were added to dry dimethylformamide (30 mL). The reaction mixture was stirred at 100 °C under inert atmosphere, until the starting material disappeared in TLC. The reaction mixture was distilled in vacuo, and ethyl acetate (100 mL) was added to the mixture. The organic extract was washed twice with saturated aqueous NaCl (20 mL), and the organic extract was dried over Na_2_SO_4_. After solvent was removed under vacuum, the product was purified as a white solid by silica gel column chromatography, 1.2 g, yield 63%. ^1^H-NMR (400 MHz, CDCl_3_): δ = 7.51–7.32 (m, 12H, H-phenyl, H-imidazolyl), 7.15–7.04 (m, 8H, H-phenyl, H-imidazolyl), 6.94 (d, *J* = 8.1 Hz, 1H, H-phenyl), 3.99 (s, 3H, CH_3_). HRMS (ESI): *m/z* [M + H]^+^ calculated for C_29_H_25_N_2_O: 417.19614; found: 417.19647.

##### Preparation of 2-(1-trityl-1*H*-imidazol-4-yl)phenol (**7**)

4-(2-Methoxyphenyl)-1-trityl-1*H*-imidazole (**6**, 1.0 g, 2.4 mmol) was added to dry dichloridemethane (20 mL), BBr_3_ (0.71 g, 2.9 mmol) was added to reaction mixture gradually at −70 °C. The reaction mixture was stirred under inert atmosphere until the starting material disappeared in TLC. The reaction mixture was diluted with dichloridemethane (50 mL), and was washed twice with saturated aqueous NaCl (20 mL), then the organic extract was dried over Na_2_SO_4_. After solvent was removed under vacuum, the product was purified as a white solid by silica gel column chromatography, 0.63 g, yield 65%. ^1^H-NMR (400 MHz, DMSO-*d*_6_): δ = 11.19 (s, 1H, OH), 7.49–7.36 (m, 2H, H-phenyl), 7.33–7.10 (m, 16H, H-phenyl, H-imidazolyl), 6.84–6.70 (m, 2H, H-phenyl), 6.45 (s, 1H, H-phenyl). ^13^C-NMR (101 MHz, DMSO-*d*_6_): δ = 147.24, 141.47, 137.70, 136.55, 128.71 (6C), 127.80 (6C), 127.56, 127.24 (3C), 126.99 (3C), 126.10, 118.39, 117.87, 115.68, 74.61. HRMS (ESI): *m/z* [M + H]^+^ calculated for C_28_H_23_N_2_O: 403.18049; found: 403.18134.

##### Preparation of 2-(3-chloropropyl)isoindoline-1,3-dione (**8**)

3-Chloropropan-1-amine (3.77 g, 40.5 mmol) and isobenzofuran-1,3-dione (2 g, 13.5 mmol) were mixed without solvent, and the reaction mixture was stirred at 140 °C (molten condition), until the starting material disappeared in TLC. The reaction mixture was distilled in vacuo, ethyl acetate (100 mL) was added to the mixture. The organic extract was washed twice with saturated aqueous NaCl (20 mL), and the organic extract was dried over Na_2_SO_4_. After solvent was removed under vacuum, the product was purified as a white solid by silica gel column chromatography, 1.66 g, yield 55%. ^1^H-NMR (400 MHz, DMSO-*d*_6_): δ = 7.85–7.78 (m, 4H, H-phenyl), 3.73–3.62 (m, 4H, CH_2_, CH_2_), 2.08–2.01(m, 2H, CH_2_). ^13^C-NMR (101 MHz, DMSO-*d*_6_): δ = 167.38 (2C), 133.75 (2C), 131.19 (2C), 122.43 (2C), 42.31, 34.60, 30.39. HRMS (ESI): *m/z* [M + H]^+^ calculated for C_11_H_11_ClNO_2_: 224.04728; found: 224.04683.

##### Preparation of 2-(3-(2-(1-trityl-1*H*-imidazol-5-yl)phenoxy)propyl)isoindoline-1,3-dione (**9**)

2-(1-Trityl-1*H*-imidazol-4-yl)phenol (**7**, 0.5 g, 1.2 mmol) and KOH (0.14 g, 2.5 mmol) were added to dry dimethylformamide (20 mL), after the reaction mixture was stirred at 20 °C for 15 min, 2-(3-chloropropyl)isoindoline-1,3-dione (**8**, 0.294 g, 1.32 mmol) was added to reaction mixture at 20 °C. After the reaction mixture was stirred at 60 °C until the starting material disappeared in thin layer chromatography (TLC), the reaction mixture was distilled in vacuo, ethyl acetate (100 mL) was added to the mixture. The organic extract was washed twice with saturated aqueous NaCl (20 mL), and the organic extract was dried over Na_2_SO_4_. After solvent was removed under vacuum, the product was purified as a white solid by silica gel column chromatography, 0.5 g, yield 71%. ^1^H-NMR (400 MHz, DMSO-*d*_6_): δ = 8.08 (dd, *J* = 7.7, 1.6 Hz, 1H, H-phenyl), 7.88–7.81 (m, 4H, H-phenyl, H-imidazolyl), 7.44–7.35 (m, 10H, H-phenyl), 7.17–7.13 (m, 7H, H-phenyl, H-imidazolyl), 7.12–7.08 (m, 1H, H-phenyl), 6.97 (t, *J* = 7.5 Hz, 1H, H-phenyl), 6.92 (d, *J* = 8.2 Hz, 1H, H-phenyl), 3.93 (t, *J* = 6.4 Hz, 2H, CH_2_), 3.43 (t, *J* = 6.5 Hz, 2H, CH_2_), 1.82–1.69 (m, 2H, CH_2_). ^13^C-NMR (101 MHz, DMSO-*d*_6_): δ = 167.25, 154.09, 141.77, 137.17, 134.72, 133.82, 131.11, 128.71 (6C), 128.65 (3C), 127.69, 127.66 (6C), 127.46, 127.36 (3C), 126.71, 125.92, 122.47, 121.86, 121.01, 120.60, 119.93, 111.33, 74.14, 64.42, 33.61, 27.54. HRMS (ESI): *m/z* [M + H]^+^ calculated for C_39_H_32_N_3_O_3_: 590.24382; found: 590.24541.

##### Preparation of 3-(2-(1-trityl-1*H*-imidazol-4-yl)phenoxy)propan-1-amine (**10**)

2-(3-(2-(1-Trityl-1*H*-imidazol-4-yl)phenoxy)propyl)isoindoline-1,3-dione (**9**, 0.4 g, 0.68 mmol) and 35% hydrazine hydrate (0.12 g, 1.35 mmol) were added to CH_3_OH, and the reaction mixture was stirred at 50 °C under inert atmosphere. After the starting material disappeared in thin layer chromatography (TLC), the reaction mixture was distilled in vacuo, ethyl acetate (100 mL) was added to the mixture. The organic extract was washed twice with saturated aqueous NaHCO_3_ and NaCl (20 mL) respectively, and the organic extract was dried over Na_2_SO_4_. After solvent was removed under vacuum, the unpurified product was added to the next step directly.

##### Preparation of 1,1,1-trifluoro-*N*-(3-(2-(1-trityl-1*H*-imidazol-4-yl)phenoxy)propyl) Methane Sulfonamide (**11**)

3-(2-(1-Trityl-1*H*-imidazol-4-yl)phenoxy)propan-1-amine (crude **10**, 0.3 g, 0.65 mmol) and *N,N*-diisopropylethylamine (0.168 g, 1.3 mmol) were added to dry dichloridemethane (20 mL), and trifluoromethanesulfonylchloride (0.12 g, 0.71 mmol) was added to reaction mixture gradually at 20 °C. After the starting material disappeared in thin layer chromatography (TLC), the reaction mixture was distilled in vacuo, the reaction mixture was diluted with dichloridemethane (50 mL), and the organic extract was washed twice with saturated aqueous NaCl (20 mL), and the organic extract was dried over Na_2_SO_4_. After solvent was removed under vacuum, the product **11** was purified as a white solid by silica gel column chromatography, 0.257 g, yield 67%. ^1^H-NMR (400 MHz, CDCl_3_): δ = 7.80 (s, 1H, NH), 7.43–7.32 (m, 10H, H-phenyl, H-imidazolyl), 7.23–7.16 (m, 8H, H-phenyl, H-imidazol), 7.07 (d, *J* = 1.5 Hz, 1H, H-phenyl), 6.97–6.86 (m, 2H, H-phenyl), 4.16 (t, *J* = 5.2 Hz, 2H, CH_2_), 3.57–3.46 (m, 2H, CH_2_), 2.19–2.06 (m, 2H, CH_2_). ^13^C-NMR (101 MHz, CDCl_3_): δ= 154.95, 142.08, 139.92, 138.22, 129.77 (6C), 128.56 (3C), 128.36, 128.15 (3C), 128.09 (6C), 121.68, 120.87, 119.04, 118.47, 111.86, 75.75, 67.83, 43.24, 29.80. HRMS (ESI): *m/z* [M + H]^+^ calculated for C_32_H_29_N_3_O_3_F_3_S: 592.18762; found: 592.18616.

##### Preparation of *N*-(3-(2-(1*H*-imidazol-5-yl)phenoxy)propyl)-1,1,1-trifluoromethane Sulfonamide (DX-03-12)

1,1,1-Trifluoro-*N*-(3-(2-(1-trityl-1*H*-imidazol-4-yl)phenoxy)propyl)methanesulfonamide (**11**, 0.2 g, 0.34 mmol) and AcOH (0.5 mL) were added to CH_3_OH (10 mL), and the reaction mixture was stirred at 20 °C. After the starting material disappeared in thin layer chromatography (TLC), the reaction mixture was distilled in vacuo, the product was purified as a white solid by silica gel column chromatography, 0.101 g, yield 86%. ^1^H-NMR (400 MHz, DMSO-*d*_6_): δ =7.99 (d, *J* = 7.4 Hz, 1H, H-phenyl), 7.72 (s, 1H, H-imidazolyl), 7.49 (s, 1H, H-imidazolyl), 7.18 (t, *J* = 7.8 Hz, 1H, H-phenyl), 7.03 (d, *J* = 8.1 Hz, 1H, H-phenyl), 6.98 (t, *J* = 7.5 Hz, 1H, H-phenyl), 4.15 (t, *J* = 6.0 Hz, 2H, CH_2_), 3.40 (t, *J* = 6.8 Hz, 2H, CH_2_), 2.06–2.09 (m, 2H, CH_2_). ^13^C-NMR (101 MHz, DMSO-*d*_6_): δ = 154.86, 135.47, 134.49, 127.46, 127.25, 124.99, 122.69, 120.97, 118.57, 112.34, 65.09, 41.39, 30.14. HRMS (ESI): *m/z* [M + H]^+^ calculated for C_13_H_15_O_3_N_3_F_3_S: 350.07807; found: 350.07782.

#### 3.1.4. Preparation of 4-chloro-2-(1-(4-(trifluoromethyl)benzyl)-1*H*-imidazol-4-yl) Phenol (DX-04-02)

##### Preparation of 4-iodo-1-(4-(trifluoromethyl)benzyl)-1*H*-imidazole (**12**)

4-Iodo-1*H*-imidazole (1 g, 5.2 mmol) and *N,N*-diisopropylethylamine (1.34 g, 10.4 mmol) were added to dry dimethylformamide (20 mL), and 1-(bromomethyl)-4-(trifluoromethyl)benzene (1.48 g, 6.2 mmol) was added to reaction mixture at 20 °C. After the reaction mixture was stirred at 20 °C until the starting material disappeared in thin layer chromatography (TLC), the reaction mixture was distilled in vacuo, dichloromethane (50 mL) was added to the mixture. The organic extract was washed twice with saturated aqueous NaHCO_3_ and NaCl (20 mL) respectively, and the organic extract was dried over Na_2_SO_4_. After solvent was removed under vacuum, the unpurified product was added to the next step directly.

##### Preparation of 4-(5-chloro-2-methoxyphenyl)-1-(4-(trifluoromethyl)benzyl)-1*H*-imidazole (DX-04-01)

4-Iodo-1-(4-(trifluoromethyl)benzyl)-1*H*-imidazole (crude **12**, 0.5 g, 1.4 mmol), (5-chloro-2-methoxyphenyl)boronic acid (0.27 g, 1.4 mmol), K_3_PO_4_ (0.9 g, 4.3 mmol), Pd(PPh_3_)_4_ (0.23 g, 0.2 mmol) were added to dry dimethylformamide (10 mL). The reaction mixture was stirred at 100 °C under inert atmosphere, until the starting material disappeared in TLC. The reaction mixture was distilled in vacuo, ethyl acetate (50 mL) was added to the mixture. The organic extract was washed twice with saturated aqueous NaCl (20 mL), and the organic extract was dried over Na_2_SO_4_. After solvent was removed under vacuum, DX-04-01 was purified as a white solid by silica gel column chromatography, 0.37 g, yield 72%. ^1^H-NMR (400 MHz, CD_3_OD): δ = 7.94 (d, *J* = 2.5 Hz, 1H, H-phenyl), 7.81 (s, 1H, H-imidazolyl), 7.65–7.63 (m, 3H, H-imidazolyl, H-phenyl), 7.40 (d, *J* = 8.0 Hz, 2H, H-phenyl), 7.15 (dd, *J* = 8.7, 2.5 Hz, 1H, H-phenyl), 6.96 (d, *J* = 8.8 Hz, 1H, H-phenyl), 5.34 (s, 2H, CH_2_), 3.86 (s, 3H, CH_3_). ^13^C-NMR (101 MHz, CD_3_OD): δ = 156.09, 142.76, 138.77, 137.13, 131.46, 129.05 (2C), 128.22, 127.44, 126.92 (2C), 124.85, 124.18, 121.78, 113.52, 112.48, 56.17, 51.22. HRMS (ESI): *m/z* [M + H]^+^ calculated forC_18_H_15_ON_2_ClF_3_: 367.08195; found: 367.08374.

##### Preparation of 4-chloro-2-(1-(4-(trifluoromethyl)benzyl)-1*H*-imidazol-4-yl)phenol (DX-04-02)

4-(5-Chloro-2-methoxyphenyl)-1-(4-(trifluoromethyl)benzyl)-1*H*-imidazole (DX-04-01, 0.3 g, 0.82 mmol) was added to dry dichloridemethane (20 mL), BBr_3_ (0.24 g, 0.98 mmol) was added to reaction mixture gradually at −70 °C. The reaction mixture was stirred under inert atmosphere until the starting material disappeared in TLC. The reaction mixture was diluted with dichloridemethane (50 mL), and was washed twice with saturated aqueous NaCl (20 mL), then the organic extract was dried over Na_2_SO_4_. After solvent was removed under vacuum, DX-04-02 was purified as a white solid by silica gel column chromatography, 0.15 g, yield 53%. ^1^H-NMR (400 MHz, CD_3_OD): δ = 7.85 (s, 1H, H-imidazolyl), 7.70–7.65 (m, 3H, H-imidazolyl, H-phenyl), 7.63 (d, *J* = 2.6 Hz, 1H, H-phenyl), 7.45 (d, *J* = 8.1 Hz, 2H, H-phenyl), 7.02 (dd, *J* = 8.7, 2.6 Hz, 1H, H-phenyl), 6.80 (d, *J* = 8.7 Hz, 1H, H-phenyl), 5.37 (s, 2H, CH_2_). ^13^C-NMR (101 MHz, CD_3_OD): δ = 155.21, 142.66, 140.49, 137.67, 129.87, 129.60, 129.12 (2C), 128.42, 126.89 (2C), 126.06, 125.01, 121.01, 118.79, 118.28, 51.17. HRMS (ESI): *m/z* [M + H]^+^ calculated forC_17_H_13_ON_2_ClF_3_: 353.06630; found: 353.06635.

#### 3.1.5. Preparation of 4-methyl-1*H*-indole-3-carboxylic Acid (DX-01-02)

Methyl 4-methyl-1*H*-indole-3-carboxylate (200 mg, 1.06 mmol) used hydrolysis reaction (2 N NaOH water solution) to afford DX-01-02 (white solid, 151 mg, yield 82%). ^1^H-NMR (400 MHz, CD_3_OD): δ = 8.18 (dd, *J* = 3.9, 1.9 Hz, 1H, H-indolyl), 7.30 (d, *J* = 8.0 Hz, 1H, H-indolyl), 7.17 (t, *J* = 7.7 Hz, 1H, H-indolyl), 7.01 (dd, *J* = 7.3, 0.7 Hz, 1H, H-indolyl), 2.77 (s, 3H, CH_3_). ^13^C-NMR (101 MHz, CD_3_OD): δ = 139.17, 133.89, 126.39, 125.96, 125.68, 120.61, 117.71, 112.19, 111.02, 23.30. HRMS (ESI): *m/z* [M + H]^+^ calculated for C_10_H_10_NO_2_: 176.07061; found: 176.07129.

#### 3.1.6. Preparation of 5-methyl-1*H*-indole-3-carboxylic Acid (DX-01-03)

Methyl 5-methyl-1*H*-indole-3-carboxylate (200 mg, 1.06 mmol) used hydrolysis reaction (2 N NaOH water solution) to afford DX-01-03 (white solid, 150 mg, yield 81%). ^1^H-NMR (400 MHz, CD_3_OD): δ = 8.15 (d, *J* = 1.8 Hz, 1H, H-indolyl), 8.07 (s, 1H, H-indolyl), 7.39 (d, *J* = 8.3 Hz, 1H, H-indolyl), 7.15 (dd, *J* = 8.3, 1.1 Hz, 1H, H-indolyl), 2.46 (s, 3H, CH_3_). ^13^C-NMR (101 MHz, CD_3_OD): δ = 137.41, 134.35, 127.80, 127.00, 122.51, 120.09, 117.20, 113.08, 110.78, 21.77. HRMS (ESI): *m/z* [M + H]^+^ calculated for C_10_H_10_NO_2_: 176.07061; found: 176.07094.

#### 3.1.7. Preparation of 5-nitro-1*H*-indole-3-carboxylic Acid (DX-01-04)

Methyl 5-nitro-1*H*-indole-3-carboxylate (200 mg, 0.91 mmol) used hydrolysis reaction (2 N NaOH water solution) to afford DX-01-04 (white solid, 172 mg, yield 92%). ^1^H-NMR (400 MHz, CD_3_OD): δ = 8.99 (d, *J* = 2.2 Hz, 1H, H-indolyl), 8.14 (s, 1H, H-indolyl), 8.12 (dd, *J* = 9.0, 2.3 Hz, 1H, H-indolyl), 7.58 (d, *J* = 9.0 Hz, 1H, H-indolyl). ^13^C-NMR (101 MHz, CD_3_OD): δ = 167.81, 144.32, 141.16, 136.51, 127.01, 118.93, 118.80, 113.45, 111.10, 49.00. HRMS (ESI): *m/z* [M + H]^+^ calculated for C_9_H_7_N_2_O_4_: 207.04003; found: 207.03940.

#### 3.1.8. Preparation of 6-nitro-1*H*-indole-3-carboxylic Acid (DX-01-05)

Methyl 6-nitro-1*H*-indole-3-carboxylate (200 mg, 0.91 mmol) used hydrolysis reaction (2 N NaOH water solution) to afford DX-01-05 (white solid, 168 mg, yield 90%). ^1^H-NMR (400 MHz, CD_3_OD): δ = 8.38 (d, *J* = 1.5 Hz, 1H, H-indolyl), 8.22 (s, 1H, H-indolyl), 8.18 (d, *J* = 8.9 Hz, 1H, H-indolyl), 8.05 (dd, *J* = 8.8, 1.7 Hz, 1H, H-indolyl). ^13^C-NMR (101 MHz, CD_3_OD): δ = 167.87, 144.97, 138.31, 136.78, 132.29, 122.19, 117.33, 109.89, 109.69. HRMS (ESI): *m/z* [M + H]^+^ calculated for C_9_H_7_N_2_O_4_: 207.04003; found: 207.03949.

#### 3.1.9. Preparation of 5-methoxy-1*H*-indole-3-carboxylic Acid (DX-01-06)

Methyl 5-methoxy-1*H*-indole-3-carboxylate (200 mg, 0.97 mmol) used hydrolysis reaction (2 N NaOH water solution) to afford DX-01-06 (white solid, 140 mg, yield 75%). ^1^H-NMR (400 MHz, CD_3_OD): δ = 7.88 (s, 1H, H-indolyl), 7.57 (d, *J* = 2.4 Hz, 1H, H-indolyl), 7.31 (d, *J* = 8.8 Hz, 1H, H-indolyl), 6.83 (dd, *J* = 8.8, 2.5 Hz, 1H, H-indolyl), 3.83 (s, 3H, CH_3_). ^13^C-NMR (101 MHz, CD_3_OD): δ = 156.99, 133.58, 128.36, 115.37, 114.23, 113.88, 113.61, 104.51, 103.64, 56.06. HRMS (ESI): *m/z* [M + H]^+^ calculated for C_10_H_10_NO_3_: 192.06552; found: 192.06645.

#### 3.1.10. Preparation of 7-bromo-1*H*-indole-3-carboxylic Acid (DX-01-07)

Methyl 7-bromo-1*H*-indole-3-carboxylate (200 mg, 0.79 mmol) used hydrolysis reaction (2 N NaOH water solution) to afford DX-01-07 (white solid, 160 mg, yield 85%). ^1^H-NMR (400 MHz, CD_3_OD): δ = 8.06 (dd, *J* = 8.0, 0.9 Hz, 1H, H-indolyl), 7.98 (s, 1H, H-indolyl), 7.36 (dd, *J* = 7.6, 0.7 Hz, 1H, H-indolyl), 7.08 (t, *J* = 7.8 Hz, 1H, H-indolyl). ^13^C-NMR (101 MHz, CD_3_OD): δ = 168.56, 138.29, 134.03, 129.07, 126.24, 123.59, 121.46, 117.02, 106.02. HRMS (ESI): *m/z* [M + H]^+^ calculated for C_9_H_7_NO_2_Br: 239.96547; found: 239.96452.

#### 3.1.11. Preparation of (*E*) 1*H*-indole-3-carbaldehyde Oxime (DX-01-13)

1*H*-indole-3-carbaldehyde (200 mg, 1.38 mmol) following the similar procedure described for the preparation of DX-02-04 afforded DX-01-13 (white solid, 158 mg, yield 72%). ^1^H-NMR (400 MHz, DMSO-*d*_6_): δ = 11.56 (s, 1H, OH), 11.17 (s, 1H, NH), 8.26 (s, 1H, =CH), 8.22 (d, *J* = 1.3 Hz, 1H, H-indolyl), 7.97 (d, *J* = 7.9 Hz, 1H, H-indolyl), 7.44 (d, *J* = 8.1 Hz, 1H, H-indolyl), 7.16 (t, *J* = 7.5 Hz, 2H, H-indolyl). ^13^C-NMR (101 MHz, CD_3_OD): δ = 146.89, 138.66, 132.19, 127.98, 123.30, 121.40, 118.79, 112.59, 107.56. HRMS (ESI): *m/z* [M + H]^+^ calculated for C_9_H_9_ON_2_: 161.07094; found: 161.07076.

#### 3.1.12. Preparation of *N*-((1*H*-indol-3-yl)methyl)hydroxylamine (DX-01-14)

DX-01-13 (100 mg, 0.62 mmol) following the similar procedure described for the preparation of DX-02-05 afforded DX-01-14 (white solid, 70 mg, yield 69%). ^1^H-NMR (400 MHz, DMSO-*d*_6_): δ = 10.87 (s, 1H, NH-indolyl), 7.58 (d, *J* = 7.8 Hz, 1H, H-indolyl), 7.34 (d, *J* = 8.1 Hz, 1H, H-indolyl), 7.29 (br, 1H, NH), 7.24 (s, 1H, H-indolyl), 7.06 (t, *J* = 8.0 Hz, 1H, H-indolyl), 6.99–6.95 (m, 1H, H-indolyl), 5.66 (br, 1H, OH), 4.04 (s, 2H, CH_2_). ^13^C-NMR (101 MHz, CD_3_OD): δ = 138.02, 128.61, 125.94, 122.58, 120.11, 119.41, 112.28, 110.24, 55.80. HRMS (ESI): *m/z* [M + H]^+^ calculated for C_9_H_11_ON_2_: 163.08659; found: 163.08611.

#### 3.1.13. Preparation of *N*-((5-(o-tolyl)-1*H*-indol-3-yl)methyl)hydroxylamine (DX-01-15)

5-Bromo-1*H*-indole-3-carbaldehyde (300 mg, 1.34 mmol) and o-tolylboronic acid (183 mg, 1.34 mmol) following the similar procedure described for the preparation of DX-03-12 and DX-02-05 afforded DX-01-15 (white solid, 145 mg, yield 43% from 5-bromo-1*H*-indole-3-carbaldehyde). ^1^H-NMR (400 MHz, CD_3_OD): δ = 7.57–7.39 (m, 3H, H-phenyl, H-indolyl), 7.26–7.04 (m, 5H, H-phenyl, H-indolyl), 5.13 (s, 2H, CH_2_), 2.24 (s, 3H, CH_3_). ^13^C-NMR (101 MHz, CD_3_OD): δ = 144.63, 137.29, 136.60, 135.38, 131.19, 131.12, 128.56, 127.79, 126.62, 125.69, 124.87, 119.47, 112.33, 107.61, 62.12, 29.31. HRMS (ESI): *m/z* [M + H]^+^ calculated for C_16_H_17_N_2_O: 253.13354; found: 253.13471.

#### 3.1.14. Preparation of (*E*)-1*H*-indole-2-carbaldehyde Oxime (DX-01-16)

1*H*-indole-2-carbaldehyde (200 mg, 1.38 mmol) following the similar procedure described for the preparation of DX-02-04 afforded DX-01-16 (white solid, 117 mg, yield 53%). ^1^H-NMR (400 MHz, DMSO-*d*_6_): δ = 11.29 (s, 1H, OH), 11.17 (s, 1H, NH), 8.14 (s, 1H, =CH), 7.52 (d, *J* = 8.0 Hz, 1H, H-indolyl), 7.37 (d, *J* = 8.2 Hz, 1H, H-indolyl), 7.12 (t, *J* = 7.6 Hz, 1H, H-indolyl), 6.98 (t, *J* = 7.5 Hz, 1H, H-indolyl), 6.56 (s, 1H, H-indolyl). ^13^C-NMR (101 MHz, CD_3_OD): δ = 142.67, 138.93, 132.77, 129.48, 124.71, 122.20, 120.89, 112.84, 108.61. HRMS (ESI): *m/z* [M + H]^+^ calculated for C_9_H_9_ON_2_: 161.07094; found: 161.07077.

#### 3.1.15. Preparation of *N*-((1*H*-indol-2-yl)methyl)hydroxylamine (DX-01-17)

DX-01-16 (60 mg, 0.37 mmol) following the similar procedure described for the preparation of DX-02-05 afforded DX-01-17 (white solid, 39 mg, yield 65%). ^1^H-NMR (400 MHz, DMSO-d_6_): δ = 10.88 (s, 1H, NH-indolyl), 7.46–7.39 (m, 2H, H-indolyl, NH), 7.32 (d, *J* = 8.3 Hz, 1H, H-indolyl), 7.03–6.97 (m, 1H, H-indolyl), 6.93 (t, *J* = 8.1 Hz, 1H), 6.28 (s, 1H, H-indolyl), 6.17 (br, 1H, OH), 4.00 (s, 2H, CH_2_). ^13^C-NMR (101 MHz, CD_3_OD): δ = 137.98, 136.56, 122.09, 120.86, 120.84, 120.01, 111.84, 101.86, 51.98. HRMS (ESI): *m/z* [M + H]^+^ calculated for C_9_H_11_ON_2_: 163.08659; found: 163.08636.

#### 3.1.16. Preparation of (*E*)-3-((1*H*-indol-3-yl)methylene)pyrrolidine-2,5-dione (DX-02-01)

1*H*-indole-3-carbaldehyde (500 mg, 3.45 mmol) and 1*H*-pyrrole-2,5-dione (401 mg, 4.14 mmol) following the similar procedure described for the preparation of DX-02-04 afforded DX-02-01 (white solid, 398 mg, yield 51% from 1*H*-indole-3-carbaldehyde). ^1^H-NMR (400 MHz, DMSO-*d*_6_): δ = 11.94 (s, 1H, NH-indolyl), 11.18 (s, 1H, NH), 7.81 (dd, *J* = 12.1, 5.1 Hz, 2H, H-indolyl, =CH), 7.70 (t, *J* = 2.1 Hz, 1H, H-indolyl), 7.54–7.41 (m, 1H, H-indolyl), 7.21–7.16 (m, 2H, H-indolyl), 3.52 (d, *J* = 2.2 Hz, 2H). HRMS (ESI): *m/z* [M + H]^+^ calculated for C_13_H_11_O_2_N_2_: 227.08150; found: 227.08134.

#### 3.1.17. Preparation of (*R/S*) 3-((1*H*-indol-3-yl)methyl)pyrrolidine-2,5-dione (DX-02-02)

DX-02-01 (200 mg, 3.45 mmol) following the similar procedure described for the preparation of DX-02-04 afforded DX-02-02 (white solid, 65 mg, yield 32%). ^1^H-NMR (400 MHz, DMSO-*d*_6_): δ = 11.06 (s, 1H, NH-indolyl), 10.91 (s, 1H, NH), 7.51 (d, *J* = 7.8 Hz, 1H, H-indolyl), 7.33 (d, *J* = 8.0 Hz, 1H, H-indolyl), 7.16 (s, 1H, H-indolyl), 7.06 (t, *J* = 7.5 Hz, 1H, H-indolyl), 6.97 (t, *J* = 7.5 Hz, 1H, H-indolyl), 3.20–3.12 (m, 1H, CH’H’’), 2.95 (dd, *J* = 14.0, 8.5 Hz, 1H, CH’H’’), 2.70–2.61 (m, 2H, H-pyrrolidine-2,5-dione), 2.34 (dd, *J* = 18.0, 3.5 Hz, 1H, H-pyrrolidine-2,5-dione). HRMS (ESI): *m/z* [M + H]^+^ calculated for C_13_H_13_N_2_O_2_: 229.09715; found: 229.09663

#### 3.1.18. Preparation of (*E*) 2-(1*H*-imidazol-4-yl)benzaldehyde Oxime (DX-03-02)

Compound **5** (400 mg, 0.92 mmol) and (2-formylphenyl)boronic acid (138 mg, 0.92 mmol) following the similar procedure described for the preparation of DX-03-12 and DX-02-04 afforded DX-03-02 (white solid, 98 mg, yield 57% from (2-formylphenyl)boronic acid). ^1^H-NMR (400 MHz, CD_3_OD): δ = 8.32 (s, 1H, =CH), 7.85–7.84 (m, 1H, H-phenyl), 7.79 (s, 1H, H-imidazolyl), 7.52 (d, *J* = 7.7 Hz, 1H, H-phenyl), 7.42–7.38 (m, 1H, H-phenyl), 7.34–7.30 (m, 1H, H-phenyl), 7.12 (s, 1H, H-imidazolyl). ^13^C-NMR (100 MHz, DMSO-*d*_6_): δ = 148.70, 136.36, 133.83, 130.10, 129.50, 129.27, 128.72, 128.34, 127.09, 126.20. HRMS (ESI): *m/z* [M + H]^+^ calculated for C_10_H_10_ON_3_: 188.08184; found: 188.08191.

#### 3.1.19. Preparation of *N*-(2-(1*H*-imidazol-4-yl)benzyl)hydroxylamine (DX-03-03)

Compound DX-03-02 (50 mg, 0.27 mmol) following the similar procedure described for the preparation of DX-02-04 afforded DX-03-03 (white solid, 32 mg, yield 63%). ^1^H-NMR (400 MHz, CD_3_OD): δ = 7.75 (s, 1H, H-imidazolyl), 7.53 (d, *J* = 1.7 Hz, 1H, H-phenyl), 7.47–7.44 (m, 1H, H-phenyl), 7.35–7.27 (m, 3H, H-imidazolyl, H-phenyl), 4.04 (s, 2H). HRMS (ESI): *m/z* [M + H]^+^ calculated for C_10_H_12_ON_3_: 190.09749; found: 190.09737.

#### 3.1.20. Preparation of (*E*) 2-(2*H*-tetrazol-5-yl)benzaldehyde Oxime (DX-03-04)

2-(2H-tetrazol-5-yl)benzaldehyde (200 mg, 1.15 mmol) following the similar procedure described for the preparation of DX-03-12 and DX-02-04 afforded DX-03-04 (white solid, 78 mg, yield 36% from 2-(2*H*-tetrazol-5-yl)benzaldehyde). ^1^H-NMR (400 MHz, DMSO-*d*_6_): δ = 11.23 (s, 1H, OH), 11.07 (s, 1H, H-tetrazolyl), 8.29 (s, 1H, =CH), 7.73–7.56 (m, 4H, H-phenyl). ^13^C-NMR (101 MHz, DMSO-*d*_6_): δ = 156.17, 137.81, 132.14, 130.64, 129.27, 127.64, 116.84, 110.69. HRMS (ESI): *m/z* [M + H]^+^ calculated for C_8_H_8_N_5_O: 190.07234; found: 190.07158.

#### 3.1.21. Preparation of (*E*) 5-chloro-2-(1*H*-imidazol-4-yl)benzaldehyde Oxime (DX-03-05)

Compound **5** (300 mg, 0.69 mmol) and (4-chloro-2-formylphenyl)boronic acid (127 mg, 0.69 mmol) following the similar procedure described for the preparation of DX-03-12 and DX-02-04 afforded DX-03-05 (white solid, 70 mg, yield 46% from (4-chloro-2-formylphenyl)boronic acid). ^1^H-NMR (400 MHz, DMSO-*d*_6_): δ = 12.38 (s, 1H, OH), 11.32 (s, 1H, NH-imidazolyl), 8.77 (s, 1H, =CH), 7.80 (s, 1H, H-imidazolyl), 7.74 (s, 1H, H-imidazolyl), 7.63 (d, *J* = 2.4 Hz, 1H, H-phenyl), 7.44–7.42 (m, 2H, H-phenyl). ^13^C-NMR (101 MHz, DMSO-*d*_6_): δ = 147.77, 138.56, 136.60, 133.12, 131.72, 131.48, 131.05, 129.25, 125.42, 116.28. HRMS (ESI): *m/z* [M + H]^+^ calculated for C_10_H_9_ClN_3_O: 222.04287; found: 222.04330.

#### 3.1.22. Preparation of (*E*) 2-fluoro-6-(1*H*-imidazol-4-yl)benzaldehyde Oxime (DX-03-06)

Compound **5** (300 mg, 0.69 mmol) and (3-fluoro-2-formylphenyl)boronic acid (116 mg, 0.69 mmol) following the similar procedure described for the preparation of DX-03-12 and DX-02-04 afforded DX-03-06 (white solid, 75 mg, yield 53% from (3-fluoro-2-formylphenyl)boronic acid). ^1^H-NMR (400 MHz, DMSO-*d*_6_): δ = 12.37 (s, 1H, OH), 11.24 (s, 1H, NH-imidazolyl), 8.37 (s, 1H, =CH), 7.76 (s, 1H, H-imidazolyl), 7.52 (s, 1H, H-imidazolyl), 7.36–7.13 (m, 3H, H-phenyl). ^13^C-NMR (101 MHz, DMSO-*d*_6_): δ = 161.48, 159.00, 144.68, 136.29, 130.27, 124.36, 122.95, 117.89, 116.35, 113.98. HRMS (ESI): *m/z* [M + H]^+^ calculated for C_10_H_9_FN_3_O: 206.07242; found: 206.07250.

#### 3.1.23. Preparation of (*E*) 2-(1*H*-imidazol-4-yl)-5-methoxybenzaldehyde Oxime (DX-03-07)

Compound **5** (300 mg, 0.69 mmol) and (2-formyl-4-methoxyphenyl)boronic acid (124 mg, 0.69 mmol) following the similar procedure described for the preparation of DX-03-12 and DX-02-04 afforded DX-03-07 (white solid, 113 mg, yield 76% from (2-formyl-4-methoxyphenyl)boronic acid). ^1^H-NMR (400 MHz, CD_3_OD): δ = 8.73 (s, 1H, H-imidazolyl), 7.78 (s, 1H, H-imidazolyl), 7.52–7.38 (m, 3H, H-phenyl), 7.15 (s, 1H, H-phenyl), 7.10–6.96 (m, 4H, H-phenyl), 3.85 (s, 3H, CH_3_). ^13^C-NMR (101 MHz, CD_3_OD): δ = 141.55, 137.87, 133.07, 132.61, 121.44, 118.60, 117.79, 116.37, 113.90, 110.95, 55.90. HRMS (ESI): *m/z* [M + H]^+^ calculated for C_11_H_12_N_3_O_2_: 218.09240; found: 218.09206.

#### 3.1.24. Preparation of (*E*) 4-(1*H*-imidazol-4-yl)-2′-(trifluoromethyl)-[1,1′-biphenyl]-3-carbaldehyde Oxime (DX-03-08)

Compound **5** (300 mg, 0.69 mmol) and (3-formyl-2′-(trifluoromethyl)-[1,1′-biphenyl]-4-yl) boronic acid (202 mg, 0.69 mmol) following the similar procedure described for the preparation of DX-03-12 and DX-02-04 afforded DX-03-08 (white solid, 82mg, yield 36% from (3-formyl-2′-(trifluoromethyl)-[1,1′-biphenyl]-4-yl)boronic acid). ^1^H-NMR (400 MHz, DMSO-*d*_6_): δ = 12.38 (s, 1H, OH), 11.14 (s, 1H, NH-imidazolyl), 8.85(s, 1H, H-imidazolyl), 7.86–7.79 (m, 2H, H-imidazolyl, H-phenyl), 7.73–7.63 (m, 4H, H-phenyl), 7.46 (br, 2H, H-phenyl), 7.33 (s, 1H, H-phenyl). ^13^C-NMR (101 MHz, DMSO-*d*_6_): δ = 148.54, 140.58, 137.83, 136.56, 132.89, 132.57, 130.11, 129.84, 129.53, 128.94, 128.64, 126.58, 126.32, 126.23, 126.04, 123.22, 116.04. HRMS (ESI): *m/z* [M + H]^+^ calculated for C_17_H_13_F_3_N_3_O: 332.10052; found: 332.10123.

#### 3.1.25. Preparation of (*E*) 3-(1*H*-imidazol-4-yl)-[1,1′-biphenyl]-4-carbaldehyde Oxime (DX-03-09)

Compound **5** (300 mg, 0.69 mmol) and (4-formyl-[1,1′-biphenyl]-3-yl)boronic acid (156 mg, 0.69 mmol) following the similar procedure described for the preparation of DX-03-12 and DX-02-04 afforded DX-03-09 (white solid, 81 mg, yield 45% from (4-formyl-[1,1′-biphenyl]-3-yl)boronic acid). ^1^H-NMR (400 MHz, DMSO-*d*_6_): δ = 11.01 (s, 1H, OH), 10.34 (s, 1H, NH-imidazolyl), 8.55 (s, 1H, =CH), 8.39 (s, 1H, H-imidazolyl), 8.12 (s, 1H, H-imidazolyl), 7.58 (br, 1H, H-phenyl), 7.48–7.22 (m, 2H, H-phenyl), 7.11–7.02 (m, 2H, H-phenyl), 6.87–6,81 (m, 2H, H-phenyl), 6.50 (s, 1H, H-phenyl). HRMS (ESI): *m/z* [M + H]^+^ calculated for C_16_H_14_N_3_O: 264.11314; found: 264.11346.

#### 3.1.26. Preparation of 4-(2-((4-(trifluoromethyl)benzyl)oxy)phenyl)-1,2,3-thiadiazole (DX-03-10)

2-(1,2,3-Thiadiazol-4-yl)phenol (100 mg, 0.56 mmol) and 1-(bromomethyl)-4-(trifluoromethyl)benzene (134 mg, 0.56 mmol) following the similar procedure described for the preparation of **12** afforded DX-03-10 (white solid, 128 mg, yield 68%). ^1^H-NMR (400 MHz, DMSO-*d*_6_): δ = 9.39 (s, 1H, H-thiadiazolyl), 8.26 (d, *J* = 7.5 Hz, 1H, H-phenyl), 7.76 (d, *J* = 8.2 Hz, 2H, H-phenyl), 7.70 (d, *J* = 8.5 Hz, 2H, H-phenyl), 7.48–7.43 (m, 1H, H-phenyl), 7.31 (d, *J* = 7.7 Hz, 1H, H-phenyl), 7.17 (t, *J* = 7.5 Hz, 1H, H-phenyl), 5.44 (s, 2H, CH_2_). HRMS (ESI): *m/z* [M + H]^+^ calculated for C_16_H_12_F_3_N_2_OS: 337.06169; found: 337.06116.

#### 3.1.27. Preparation of *N*-(3-(2-(1*H*-imidazol-5-yl)phenoxy)propyl)-aminesulfonamide (DX-03-11)

Compound **9** (200 mg, 0.34 mmol) following the similar procedure described for the preparation of DX-03-12 afforded DX-03-11 (white solid, 45 mg, yield 45% from compound **9**). ^1^H-NMR (400 MHz, CD_3_OD): δ = 8.67 (s, 1H, NH-imidazolyl), 8.08 (s, 1H, H-imidazolyl), 7.83 (s, 1H, H-imidazolyl), 7.69 (d, *J* = 7.7 Hz, 1H, H-phenyl), 7.40 (t, *J* = 7.1 Hz, 1H, H-phenyl), 7.16 (d, *J* = 8.2 Hz, 1H, H-phenyl), 7.08 (t, *J* = 7.4 Hz, 1H, H-phenyl), 4.20 (t, *J* = 6.0 Hz, 2H, CH_2_), 3.44 (t, *J* = 6.9 Hz, 2H, CH_2_), 2.11–2.03 (m, 2H, CH_2_). HRMS (ESI): *m/z* [M + H]^+^ calculated for C_12_H_17_O_3_N_4_S: 297.10159; found: 297.10115.

#### 3.1.28. Preparation of 1-(3-(2-(1*H*-imidazol-5-yl)phenoxy)propyl)-3-(4-(trifluoromethyl) phenyl)urea (DX-03-13)

Compound **9** (200 mg, 0.34 mmol) following the similar procedure described for the preparation of DX-03-12 afforded DX-03-13 (white solid, 77 mg, yield 56% from compound **9**). ^1^H-NMR (400 MHz, DMSO-*d*_6_): δ = 12.24 (s, 1H, NH-imidazolyl), 8.90 (s, 1H, H-imidazolyl), 8.01 (d, *J* = 7.4 Hz, 1H, H-phenyl), 7.75 (s, 1H, H-imidazolyl), 7.61–7.54 (m, 5H, H-phenyl), 7.21–7.15 (m, 1H, H-phenyl), 7.06 (d, *J* = 7.8 Hz, 1H, H-phenyl), 7.02–6.94 (m, 1H, H-phenyl), 6.48 (t, *J* = 5.7 Hz, 1H, H-phenyl), 4.14 (t, *J* = 6.1 Hz, 2H, CH_2_), 3.38–3.32 (m, 2H, CH_2_), 2.08–1.97 (m, 2H, CH_2_). ^13^C-NMR (101 MHz, DMSO-*d*_6_): δ = 155.42, 155.08, 144.71, 135.42, 129.17, 127.38, 127.13, 126.38 (2C), 123.78, 122.81, 121.53, 121.21, 120.86, 117.71 (2C), 112.33, 65.82, 36.89, 29.98. HRMS (ESI): *m/z* [M + H]^+^ calculated for C_20_H_20_O_2_N_4_F_3_: 405.15329; found: 405.15302.

#### 3.1.29. Preparation of 1-(3-bromo-4-fluorobenzyl)-4-(5-chloro-2-methoxyphenyl)-1*H*-imidazole (DX-04-03)

2-Bromo-4-(bromomethyl)-1-fluorobenzene (500 mg, 1.88 mmol) following the similar procedure described for the preparation of DX-04-02 afforded DX-04-03 (white solid, 480 mg, yield 65% from 2-bromo-4-(bromomethyl)-1-fluorobenzene). ^1^H-NMR (400 MHz, CD_3_OD): δ = 7.95 (d, *J* = 2.7 Hz, 1H, H-phenyl), 7.78 (d, *J* = 1.3 Hz, 1H, H-imidazolyl), 7.64 (d, *J* = 1.3 Hz, 1H, H-imidazolyl), 7.57 (dd, *J* = 6.5, 2.2 Hz, 1H, H-phenyl), 7.30–7.24 (m, 1H, H-phenyl), 7.20 (d, *J* = 8.5 Hz, 1H, H-phenyl), 7.16 (dd, *J* = 8.8, 2.7 Hz, 1H, H-phenyl), 6.98 (d, *J* = 8.8 Hz, 1H, H-phenyl), 5.22 (s, 2H, CH_2_), 3.88 (s, 3H, CH_3_). ^13^C-NMR (101 MHz, CD_3_OD): δ = 161.35, 158.96, 156.10, 138.33, 137.40, 133.91, 129.70, 128.04, 127.36, 126.81, 125.19, 121.53, 117.84, 113.47, 110.01, 56.13, 50.30. HRMS (ESI): *m/z* [M + H]^+^ calculated for C_17_H_14_ON_2_BrClF: 394.99566; found: 394.99728.

#### 3.1.30. Preparation of 2-(1-(3-bromo-4-fluorobenzyl)-1*H*-imidazol-4-yl)-4-chlorophenol (DX-04-04)

DX-04-03 (200 mg, 0.51 mmol) following the similar procedure described for the preparation of DX-04-02 afforded DX-04-04 (white solid, 116 mg, yield 60%). ^1^H-NMR (400 MHz, DMSO-*d*_6_): δ = 9.12 (s, 1H, H-imidazolyl), 8.40 (s, 1H, H-imidazolyl), 8.01 (s, 1H, H-imidazolyl), 7.89 (dd, *J* = 6.5, 1.7 Hz, 1H, H-phenyl), 7.56–7.49 (m, 1H, H-phenyl), 7.45 (t, *J* = 8.6 Hz, 1H, H-phenyl), 7.34 (dd, *J* = 8.8, 2.5 Hz, 1H, H-phenyl), 7.00 (d, *J* = 8.8 Hz, 1H, H-phenyl), 5.42 (s, 2H, CH_2_). ^13^C-NMR (101 MHz, CD_3_OD): δ = 161.85, 159.35, 154.93, 136.62, 134.79, 130.61, 130.34, 126.90, 125.53, 119.70, 118.77, 118.32, 118.09, 117.92, 110.27, 51.71. HRMS (ESI): *m/z* [M + H]^+^ calculated for C_16_H_12_ON_2_BrClF: 380.98001; found: 380.98065.

#### 3.1.31. Preparation of 4-chloro-2-(1-(4-(trifluoromethyl)benzyl)-1*H*-1,2,3-triazol-4-yl)phenol (DX-04-05)

DX-04-06 (300 mg, 0.81 mmol) following the similar procedure described for the preparation of DX-04-02 afforded DX-04-05 (white solid, 176 mg, yield 61%). ^1^H-NMR (400 MHz, DMSO-*d*_6_): δ = 10.49 (s, 1H, OH), 7.85 (s, 1H, H-triazolyl), 7.64 (d, *J* = 8.1 Hz, 2H, H-phenyl), 7.34 (dd, *J* = 8.8, 2.7 Hz, 1H, H-phenyl), 7.20 (d, *J* = 8.1 Hz, 2H, H-phenyl), 7.16 (d, *J* = 2.7 Hz, 1H, H-phenyl), 6.98 (d, *J* = 8.8 Hz, 1H, H-phenyl), 5.62 (s, 2H, CH_2_). ^13^C-NMR (101 MHz, CD_3_OD): δ = 155.39, 140.99, 136.49, 134.75, 132.46, 131.76, 131.11, 129.36 (2C), 126.49 (2C), 125.38, 124.09, 118.30, 116.15, 53.27. HRMS (ESI): *m/z* [M + H]^+^ calculated for C_16_H_12_ON_3_ClF_3_: 354.06155; found: 354.06015.

#### 3.1.32. Preparation of 4-(5-chloro-2-methoxyphenyl)-1-(4-(trifluoromethyl)benzyl)-1*H*-1,2,3-triazole (DX-04-06)

4-chloro-2-ethynyl-1-methoxybenzene (500 mg, 3.01 mmol) and 1-(bromomethyl)-4-(trifluoromethyl)benzene (717 mg, 3.01 mmol) following the similar procedure described for the preparation of DX-04-02 afforded DX-04-06 (white solid, 675 mg, yield 61% from 4-chloro-2-ethynyl-1-methoxybenzene). ^1^H-NMR (400 MHz, CD_3_OD): δ = 7.71 (s, 1H, H-triazolyl), 7.50 (d, *J* = 8.1 Hz, 2H, H-phenyl), 7.40 (dd, *J* = 8.9, 2.7 Hz, 1H, H-phenyl), 7.14 (d, *J* = 8.1 Hz, 2H, H-phenyl), 7.09 (d, *J* = 2.6 Hz, 1H, H-phenyl), 7.03 (d, *J* = 8.9 Hz, 1H, H-phenyl), 5.55 (s, 2H, CH_2_), 3.64 (s, 3H, CH_3_). ^13^C-NMR (101 MHz, CD_3_OD): δ = 154.74, 143.61, 131.20, 130.91, 129.93 (2C), 129.81, 129.39, 127.59, 127.02 (2C), 125.98, 125.64, 118.63, 117.43, 55.31, 55.04. HRMS (ESI): *m/z* [M + H]^+^ calculated for C_17_H_14_ON_3_ClF_3_: 368.07720; found: 368.07599.

#### 3.1.33. Preparation of 4-chloro-2-(5-(4-(trifluoromethyl)phenyl)-1*H*-1,2,3-triazol-4-yl)phenol (DX-04-07)

DX-04-08 (200 mg, 0.57 mmol) following the similar procedure described for the preparation of DX-04-02 afforded DX-04-07 (white solid, 144 mg, yield 75%). ^1^H-NMR (400 MHz, DMSO-*d*_6_): δ = 15.28 (s, 1H, H-triazolyl), 9.91 (s, 1H, NH-triazolyl), 7.75–7.68 (m, 4H, H-phenyl), 7.37 (br, 2H, H-phenyl), 6.94 (s, 1H, H-phenyl). ^13^C-NMR (101 MHz, CD_3_OD): δ = 155.58, 142.95, 137.77, 136.14, 131.62, 131.27, 130.86, 128.63 (2C), 126.43 (2C), 125.31, 124.28, 119.14, 118.53. HRMS (ESI): *m/z* [M + H]^+^ calculated for C_15_H_10_ON_3_ClF_3_: 340.04590; found: 340.04453.

#### 3.1.34. Preparation of 4-(5-chloro-2-methoxyphenyl)-5-(4-(trifluoromethyl)phenyl)-1*H*-1,2,3-triazole (DX-04-08)

4-chloro-2-ethynyl-1-methoxybenzene (300 mg, 1.81 mmol) and 1-iodo-4-(trifluoromethyl)benzene (491 mg, 1.81 mmol) following the similar procedure described for the preparation of DX-04-02 afforded DX-04-08 (white solid, 338 mg, yield 53% from 4-chloro-2-ethynyl-1-methoxybenzene). ^1^H-NMR (400 MHz, DMSO-*d*_6_): δ = 15.41 (s, 1H, NH-triazolyl), 7.74 (d, *J* = 8.1 Hz, 2H, H-phenyl), 7.68–7.59 (m, 2H, H-phenyl), 7.54 (d, *J* = 8.9 Hz, 1H, H-phenyl), 7.48 (d, *J* = 2.7 Hz, 1H, H-phenyl), 7.15 (s, 1H, H-phenyl), 3.42 (s, 3H, CH_3_). ^13^C-NMR (101 MHz, CD_3_OD): δ = 157.20, 136.42, 131.71, 131.59, 131.14, 129.65, 128.37 (2C), 126.96, 126.75, 126.40 (2C), 124.27, 121.57, 114.17, 55.99. HRMS (ESI): *m/z* [M + H]^+^ calculated for C_16_H_12_ON_3_ClF_3_: 354.06155; found: 354.06079.

### 3.2. Pharmacological Method

#### 3.2.1. Enzyme-Based IDO1s or TDOs Activity Assay 

The compounds on IDO1 or TDO activity were determined as follows [16,24,25]: A standard reaction mixture (100 μL) containing 100 mM potassium phosphate buffer (pH 6.5), 40 mM ascorbic acid (neutralized with NaOH), 200 μg/mL catalase, 20 μM methylene blue and 0.05 μM rhIDO1 or rhTDO was added to the solution containing the substrate l-tryptophan and the test sample at a determined concentration. The reaction was carried out at 37 °C for 45 min and stopped by adding 20 μL of 30% (*w/v*) trichloroacetic acid. After heating at 65 °C for 15 min, 100 μL of 2% (*w/v*) p-dimethylaminobenzaldehyde in acetic acid was added to each well. The yellow pigment derived from kynurenine was measured at 492 nm using a SYNERGY-H1 microplate reader (Biotek Instruments, Inc., Winooski, VT, USA). IC_50_ was analyzed using the GraphPad Prism 8.0 software (GraphPad Software, San Diego, CA, USA).

#### 3.2.2. Cell Culture

B16-F10 mouse melanoma cells, NCI-H460 human lung cancer cell line and MCF7 were obtained from American Type Culture Collection (ATCC). All the cell lines were cultured with Dulbecco’ modified Eagle’s medium (DMEM) (Invitrogen Corporation, Waltham, MA, USA) supplemented with 10% fetal bovine serum (FBS) (Hyclone, Waltham, MA, USA) and 1% penicillin streptomycin (Invitrogen) at 37 °C in 5% CO_2_.

#### 3.2.3. Cell Viability Assay

The in vitro inhibitory effects of the compounds were measured by MTT assay [26]. Briefly, different kinds of cells were plated in 96-well plates at a density of 2000/well. After attachment for overnight at 37 °C, the cells were treated with the compounds at various concentrations. After 72 h culture, the MTT reagent was added to each well, and plates were incubated for another 4 h at 37 °C. Then, the blue-purple crystal formamidine was dissolved in DMSO after discarding the supernatant. Samples were measured using microplate reader at a wavelength of 570 nm (Biotek Instruments, Inc., Winooski, VT, USA). IC_50_ values were calculated with GraphPad Prism 8.0 software.

#### 3.2.4. Mice

C57BL/6 mice were obtained from Beijing Vital River Laboratory Animal Technology Co., Ltd. (Beijing Vital River Laboratory Animal Technology Co., Ltd, Beijing, China). Studies involving mice were approved by the Experimental Animal Management and Welfare Committee at the Institute of Materia Medica, Peking Union Medical College.

#### 3.2.5. Pharmacokinetic Studies

The animal Care and Welfare Committee of Institute of Materia Medica, Chinese Academy of Medical Sciences approved all animal care, housing, and laboratory procedures. Male ICR mice were used in the single dose pharmacokinetic studies. Compound DX-03-12 was prepared as 3 mg/mL suspension with 0.5% CMC (containing 0.3% Tween 80) for oral use and was formulated as 3 mg/mL solution with 5% DMSO in saline for intravenous injection. Twenty mice were divided into two groups, 10 in each group. After fasting 12 h with free access to water, mice were treated with a 3 mg/kg i.v. or 30 mg/kg oral dose of compound DX-03-12. Blood samples (50 µL) were collected at 5, 15, 30 min, 1, 2, 4, 6, 8, 12 and 24 h after oral administration and 2, 5, 15, 30 min, 1, 2, 4, 6, 8, 12 and 24 h after intravenous injection. After centrifugation, the plasma samples (15 µL) were precipitated by five volumes of acetonitrile. The supernatant were analyzed by liquid chromatography/tandem mass spectrometry (Agilent Technologies, Santa Clara, CA, USA) with a Zobax C18 column (50 mm × 2.1 mm, 3.5 µm). Compound detection was performed with the mass spectrometer in MRM (multiple reaction monitoring, Agilent Technologies, Santa Clara, CA, USA) negative ionization mode. The selected reaction monitoring transitions were *m/z* 348.5→160 for compound DX-03-12. The pharmacokinetic parameters were calculated with WinNonlin software V6.3 using non-compartmental analysis (Pharsight Corporation, Mountain View, CA, USA).

#### 3.2.6. In Vivo Studies

The mouse melanoma cells B16F10 were cultured and harvested in saline. At day 0 of the experiment, 1 × 10^6^ cells were injected subcutaneously into mice, and treatment was initiated at day 1 following the mice enrolled randomly in control and experimental groups [27]. For control group, 0.5% CMCNa was orally administered every day. Compound DX-03-12 was dissolved in 0.5% CMC for oral treatment or dissolved in saline for intraperitoneal treatment. When the mice were sacrificed, the tumors were stripped and weighted. The tumor growth inhibition (TGI) was calculated as TGI = (1 – tumor weight _treatment_/tumor weight _vehicle_) × 100%. The statistical analysis was performed with GraphPad Prism 8.0 software and the significance level was evaluated with a one-way ANOVA model.

## 4. Conclusions

We designed four series of IDO1 inhibitors using indole and phenylimidazole as the scaffolds. The design strategies were based on the mechanism of IDO1, focusing on the interaction between the inhibitor and the heme iron. The four series of compounds were synthesized and their IDO1 and TDO inhibition was evaluated. Most compounds showed marginal IDO1 inhibition, but eight compounds showed moderate IDO1 inhibition. In particular, DX-03-12 and DX-03-13 showed IDO1 inhibition with IC_50_s of 0.3 and 0.5 μM and low cell cytotoxicities against two cancer cell lines. A pharmacokinetic study showed that DX-03-12 had satisfactory properties, with rapid absorption, moderate plasma clearance, acceptable half-life, and high oral bioavailability. The in vivo anti-tumor efficacy in a B16F10 subcutaneous xenograft mouse model showed that DX-03-12 orally administered at a dose of 60 mg/kg inhibited tumor growth by 72.2% compared with the control tumors. In summary, IDO1 inhibitor DX-03-12 exhibits promising drug like properties, good pharmacokinetic properties, low cellular toxicity, and impressive in vivo anti-tumor efficacy.

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
