# Peer review of "Design and Synthesis of Indoleamine 2,3-Dioxygenase 1 Inhibitors and Evaluation of Their Use as Anti-Tumor Agents"

_molecules, 2019, doi:10.3390/molecules24112124_

Round 1

Reviewer 1 Report

In this paper the authors report the design and synthesis of IDO 1 inhibitors and evaluation as anti tumor agents. This enzyme is an interesting target to block tumor immune escape. Accordingly, the development of potent inhibitors is of relevance in the field of cancer immunotherapy. The introduction is well presented and properly referenced. Given the interest for medicinal chemists in the field of cancer research, we would recommend publication subjected to the following revisions:

The design of the compounds should be improved in the figure, and highlighting the active site of the enzyme from PDB data would add value to this section. Also, indication of which group selected for the red star and blue rectangle should be anticipated and rationalized. For instance, why hydroxylamine and sulfoxide groups were selected?

Schemes need revisions:

Scheme 1: pyrrole dione is reported in the scheme, so there is no need to report it in reaction conditions; Reaction yields should be reported in Scheme 1, 2 and 3; Scheme 3 lacks trityl chloride;

The results and discussion with respect to bioassay data is difficult to follow, as poor SAR and discussion of the hits among each group is not discussed in detail. Also, for the lead compound DX-03-12 some considerations about the structure-activity relationship should be outlined, with respect to reference bioactive compounds.

In the experimental section several intermediates lack 13C NMR data, which are required for purity and identity as a standard in the characterization of newly synthesized compounds. Thus, they should be added in a revised version.

Minor: replace nm with nM

Author Response

Dear Reviewer:

Thanks for all of the concerns and suggestion. Please see the revised version and the response letter for all of the revision.

Best,

Huaqing Cui

Reviewer 2 Report

This paper fits the scope of the journal since it reports the design, synthesis and of biological activities of new indoleamine 2,3-dioxygenase 1 inhibitors as antitumor agents. The manuscript appears well-written and composed in a clear fashion style. In general, the procedures employed seem rational. Also the experimental section is enough detailed.

However, there are several issues with the manuscript, which lead me to conclude that this work is not suitable for publication in Molecules in its current form and some changes would need to be implemented:

1. Introduction

I recommend updating the references used for the Introduction. Specifically, from the reference 4 to 6 (Chin Med J (Engl) 2009, 122 (24), 3072-7, Curr Opin Immunol  2006, 18 (2), 220-5, J Clin Invest 2007, 117 (5), 1147-54).

In line 61 the IC50 of PF-0684003 is reported as 450 nm but in Figure 1 is indicated as 410 nM. This mistake must be corrected.

In line 78 the reference 18 appears in superscript and the others in square brackets. This mistake must be corrected.

2. Results and Discussion

2.1. Design strategy for the four compound series

On page 2 line 78 the reference 18 appears in superscript and the others in square brackets. This mistake must be corrected.

Series 3 In Figure 2 should be Series 4 since the structure of the DX-04 compounds correspond to Series 3 and Series 4 should be Series 3 (the structures of the DX-03 compounds correspond to Series 4).

2.3. In vitro biological evaluation

In line 152 after “Some commercially available compounds were purchase” the authors should indicate “(see Section 3.1)”, for a better understanding of the article.

In Table 1.2 3 and 4:

a.    The hyphen indicating the non-detectable inhibition of compounds does not appear.

b.    The authors should mention why they have used NLG919 and Taxol as reference compounds.

c.     The size of all structures should be the same.

In line 197 “(DX-03-01 to DX-03-10)” should be change by “(DX-03-01 to DX-03-09)”.

All the cytotoxicities againts H460 and MCF7 of Table 4 should include the units (% or μM).

2.4. Pharmacokinetic study of DX-03-12

In line 220, the size of the letter is larger. The same size should be used thorough the manuscript.

In Figure 3 the abbreviations PO and iv are used. The meaning of these abbreviations should be indicated either in the figure legend or in the text.

2.4. In vitro anti-tumor efficacy study of compound DX-03-12

In line 245 shows the same numbering as the previous section 2.4. and it should be 2.5. Consequently the paragraph of conclusions would be 2.6.

In Figure 4. A, the authors should explain what is CTX and why they have used it in the experiment.

2.5. Conclusions

The compounds DX-03-12 and DX-03.13 show low cell cytotoxicities against two cancer cell lines and the authors suggest that they is immune modulators. I recommend carrying out studies that demonstrate this mechanism of action.

3. Experiment Section

3.1. Chemistry       

From line 332 to 357 and from line 411 to 420 the line spacing is different. The same format should be used thorough the manuscript.

In line 443 the number of compound 8 should be bold font. This mistake must be corrected.

In the preparation of compounds DX-01-02, DX-02-03, DX-02-04, DX-02-05, DX-02-06, DX-02-07 the reaction conditions are not described and no references have been used.

In line 596, 605, 623, 639, 658 compound DX-02-04 should be change by DX-02-05.

In line 723 compound DX-04-02 should be change by 12.

Author Response

(The authors gave the same response as above.)

Reviewer 3 Report

IDO1 is an important immune chekpoint molecule that represents an important drug target in cancer immunotherapy. In the current manuscript, Wen et al aimed at identifying novel selective IDO1 catalytic inhibitors with good PK and efficacy in a mouse tumor model. They chose to develop 4 series of compounds, two starting from an indole moiety and the other two from phenylimidazole. They found, among one of the series developed from phenylimidazole, two interesting compounds inhibiting IDO1 but not TDO with no significant cytotoxicity. One of these compounds was further tested for PK by the oral and i.v. route and for in vivo efficacy in mice injected s.c. with B16 melanoma cells. They found that the examined compound displayed very good PK, with a rather long blood t1/2 and excellent bioavailability, and, when daily administered either i.p. or orally starting from one day after tumor injection, exhibited significant therapeutic effects. I have just some minor concerns for this paper as outlined below.

1) In the Introduction and also in a subsequent part of the paper, the Authors mentioned N-formylkynurenine (the direct product of IDO1) as the molecule responsible for the immunoregulatory effects of IDO1. However, this is not correct. L-Kynurenine and other downstream kynurenines (3-HK and 3-HAA) and not formyl-kynurenine have been found to mediate a great portion of the immunoregulatory effects of IDO1, which comprise Trp starvation (and thus inhibition of the proliferation of effector T cells), induction of apoptosis (mainly 3-HK and 3-HAA on some subsets of T helper cells) and conversion of effector T cells into regulatory Foxp3+ T cells (Treg). L-kynurenine induces its immunoreuglatory effects by activating the arylhydrocarbon receptor.

2) Results line 192: the phrase ...because they are IDO1 inhibitors, these compounds are not toxic to mammalian cancer cell lines and may be immune modulators: the phrase is too weak and speculative

3) Fig. 4: please specify what CTX is (the readers of Molecules may not be familiar with anti-cancer drugs); results should also be shown as tumor volume over time. In fact, it is not clear, for example, whether the compound also delays the growth of the tumor (an indication that could be very important). Assessing the drug effects by sacrificing mice at only one day does not provide a full picture of the drug effect. If the Authors wanted to demonstrate an immunoregulatory effect of their compound, they should at least  measure Foxp3 expression in CD4 CD25 cells in draining lymph nodes. It would be helpful to couple a post-hoc Bonferroni statistical analysis to the ANOVA analysis. In this way, they could also verify whether there is a significant dose-dependency and whether groups treated with the novel compounds exhibit a tumor growth comparable with that observed in mice treated with CTX (should be not significant)

Author Response

(The authors gave the same response as above.)

Reviewer 4 Report

It is well known that IDO are important target for anti-cancer drug. Therefore, development competition for the IDO inhibitors is in progress. The authors conducted research aimed at rational development of IDO inhibitors.

As a result, a compound DX-03-12 showing anti-tumor activity by oral administration was found.

Since DX-03-12 is a small compound, further improvement is also possible in the future.

Minor points:

1)    Following reference should be cited, because the DX-03-13 is similar to known IDO inhibitor in structure.

WO 2011056652 A1 20110512

2)    Iron atom should be drawn in figure 2.

3)    262: It is strange “The is also...”

Author Response

(The authors gave the same response as above.)

Round 2

Reviewer 1 Report

The manuscript has been improved according to reviewer's comments, thus we recommend publication of the work in this revised form.

nb. NA = not tested

This manuscript is a resubmission of an earlier submission. The following is a list of the peer review reports and author responses from that submission.